# A cell autonomous torsinA requirement for cholinergic neuron survival and motor control

Samuel S Pappas[1], Jay Li[1,2], Tessa M LeWitt[1], Jeong-Ki Kim[3,4,5], Umrao R Monani[3,4,5], William T Dauer[1,2,6]*

[1]Department of Neurology, University of Michigan, Ann Arbor, United States; [2]Cell and Molecular Biology Program, University of Michigan, Ann Arbor, United States; [3]Department of Cell Biology, Columbia University Medical Center, New York, United States; [4]Center for Motor Neuron Biology and Disease, Columbia University Medical Center, New York, United States; [5]Department of Pathology, Columbia University Medical Center, New York, United States; [6]Department of Cell and Developmental Biology, University of Michigan, Ann Arbor, United States

**Abstract** Cholinergic dysfunction is strongly implicated in dystonia pathophysiology. Previously (Pappas et al., 2015;4:e08352), we reported that Dlx5/6-Cre mediated forebrain deletion of the DYT1 dystonia protein torsinA (Dlx-CKO) causes abnormal twisting and selective degeneration of dorsal striatal cholinergic interneurons (ChI) (Pappas et al., 2015). A central question raised by that work is whether the ChI loss is cell autonomous or requires torsinA loss from neurons synaptically connected to ChIs. Here, we addressed this question by using ChAT-Cre mice to conditionally delete torsinA from cholinergic neurons ('ChAT-CKO'). ChAT-CKO mice phenocopy the Dlx-CKO phenotype of selective dorsal striatal ChI loss and identify an essential requirement for torsinA in brainstem and spinal cholinergic neurons. ChAT-CKO mice are tremulous, weak, and exhibit trunk twisting and postural abnormalities. These findings are the first to demonstrate a cell autonomous requirement for torsinA in specific populations of cholinergic neurons, strengthening the connection between torsinA, cholinergic dysfunction and dystonia pathophysiology.
DOI: https://doi.org/10.7554/eLife.36691.001

*For correspondence:
dauer@med.umich.edu

Competing interests: The authors declare that no competing interests exist.

## Introduction

Multiple lines of evidence implicate striatal cholinergic dysfunction in dystonia pathophysiology (*Pappas et al., 2015*; *Albin et al., 2003*; *Eskow Jaunarajs et al., 2015*; *Pappas et al., 2014*). The symptoms of DYT1 dystonia, caused by a loss of function mutation in the gene encoding torsinA (*Ozelius et al., 1997*), are reduced by antimuscarinic treatments (*e.g.,* trihexyphenidyl)(*Burke et al., 1986*). Antimuscarinic agents also reduce motor (*Pappas et al., 2015*) and electrophysiological (*Maltese et al., 2014*) abnormalities in DYT1 mouse models. Striatal cholinergic dysfunction is a common feature of multiple DYT1 animal models (*Pappas et al., 2015*; *Martella et al., 2009*; *Pisani et al., 2006*; *Sciamanna et al., 2012a*; *Sciamanna et al., 2012b*), and experimental ablation of striatal cholinergic interneurons (ChI) can lead to abnormal postures (*Kaneko et al., 2000*).

We demonstrated previously that deletion of torsinA from forebrain GABAergic and cholinergic neurons (using Dlx5/6-cre; 'Dlx-CKO') causes highly selective degeneration of dorsal striatal ChI roughly coincident with the juvenile onset of abnormal limb clasping and twisting movements (*Pappas et al., 2015*). Selective ChI abnormalities are also present in postmortem tissue from DYT1 subjects (*Pappas et al., 2015*). Abnormal movements in Dlx-CKO mice are reduced by clinically relevant antimuscarinic treatments, strengthening model therapeutic validity and suggesting shared

pathophysiology with human dystonia. This work highlights the importance of elucidating the mechanism of selective ChI loss. A critical first step toward this goal is to determine whether the ChI loss observed in Dlx-CKO mice results from a cell autonomous role of torsinA in these cells or, alternatively, whether loss of torsinA from synaptically connected cells is also required. The major aim of these studies was to address this fundamental question.

To determine whether torsinA-related ChI loss is cell autonomous, we generated and characterized cholinergic neuron selective conditional torsinA knockout mice (ChAT-CKO). We find that ChAT-CKO mice phenocopy the selective degeneration of dorsal striatal ChI observed in Dlx-CKO mice (basal forebrain neuron numbers are normal in both models). Assessment of non-forebrain cholinergic populations demonstrates that pedunculopontine and laterodorsal tegmental brainstem cholinergic neurons, and spinal motor neurons also require torsinA for survival or normal function. ChAT-CKO mice exhibit severe motor and postural abnormalities that are distinct from Dlx-CKO mice. These findings are the first to establish a cell autonomous requirement for torsinA in ChI, as well as identifying additional vulnerable cholinergic neuron populations. This *in vivo* study fundamentally advances and expands understanding of the requirement of torsinA for normal cholinergic system function, opening new directions for the study of mechanisms contributing to selective neuronal dysfunction in dystonia.

## Results and discussion

To determine if ChI neurodegeneration is a cell autonomous effect of torsinA loss, we conditionally deleted torsinA from cholinergic neurons (*Chat-IRES-Cre*[+], *Tor1a*[Flx/-]; 'ChAT-CKO' mice; Cre-recombinase expression occurs before birth and is completely selective for cholinergic neurons; *Figure 1— figure supplement 1* [*Madisen et al., 2010*]). Unilateral unbiased stereology of ChAT-immunoreactive neurons in the dorsal striatum from 1 year old mice demonstrates a ~ 34% reduction in the number of dorsal striatal ChI in ChAT-CKO mice compared to control mice (*Figure 1A,B*). This finding was confirmed in an independent cohort using bilateral unbiased stereology (48% reduction; *Figure 1—figure supplement 2A*). The number of striatal non-cholinergic neurons was not different from controls (*Figure 1—figure supplement 2B,C*), demonstrating that there are no secondary cell loss effects of ChI degeneration, and that torsinA loss of function-mediated neurodegeneration is highly specific. These findings establish a cell autonomous torsinA requirement for ChI survival.

ChI cell loss is strikingly selective in Dlx-CKO mice, occurring primarily in the dorsal aspects of the striatum, with approximately six times greater cell loss in the dorsolateral compared to ventromedial striatum (57% vs 9% cell density reduction in Dlx-CKO mice; [*Pappas et al., 2015*]). To examine if the cell autonomous ChI degeneration in ChAT-CKO mice follows a similar subregion-selective pattern, we determined the density of ChAT-immunoreactive neurons in each quadrant of the dorsal striatum (as previously [*Pappas et al., 2015*]). Significant reductions in ChI number were limited to the dorsolateral and dorsomedial segments of the dorsal striatum (72% and 54% cell density reductions in dorsolateral and dorsomedial, vs 12% and −4% in ventrolateral and ventromedial segments; *Figure 1C*). This topographic pattern of cell loss was present throughout the entire rostro-caudal extent of the striatum (*Figure 1C,D*, *Figure 1—figure supplement 3*). The dorsolateral selectivity of ChI neuron loss is highly relevant, as the dorsolateral striatum is a key motor circuit node functionally integrated according to topographic inputs, whereas ventromedial striatal neurons are connected in associative and limbic circuits (*Alexander et al., 1986*; *Haber, 2016*; *Parent and Hazrati, 1995*). In contrast, the basal forebrain contains cholinergic projection neurons subserving cognitive and attentional control (*Hasselmo and Sarter, 2011*; *Ballinger et al., 2016*), which do not degenerate in <u>either</u> model (*Figure 1E,F*). Conditional deletion of torsinA from forebrain cholinergic neurons therefore mimics the region-selective vulnerability observed in Dlx-CKO mice, demonstrating a cell autonomous requirement for torsinA in select cholinergic populations. To determine if differing time courses of torsinA loss (via differing torsinA half lives) contributes to selective vulnerability, we assessed torsinA levels in dorsal vs ventral striatal ChI at P0. Surprisingly, despite uniform prenatal Cre recombinase expression and preferential loss of dorsal ChI, torsinA levels were reduced to a greater extent in ventral ChI (dorsal ChI contained 82% of control torsinA levels, while ventral ChI had ~52% remaining; *Figure 1—figure supplement 4*). Non-vulnerable basal forebrain cholinergic

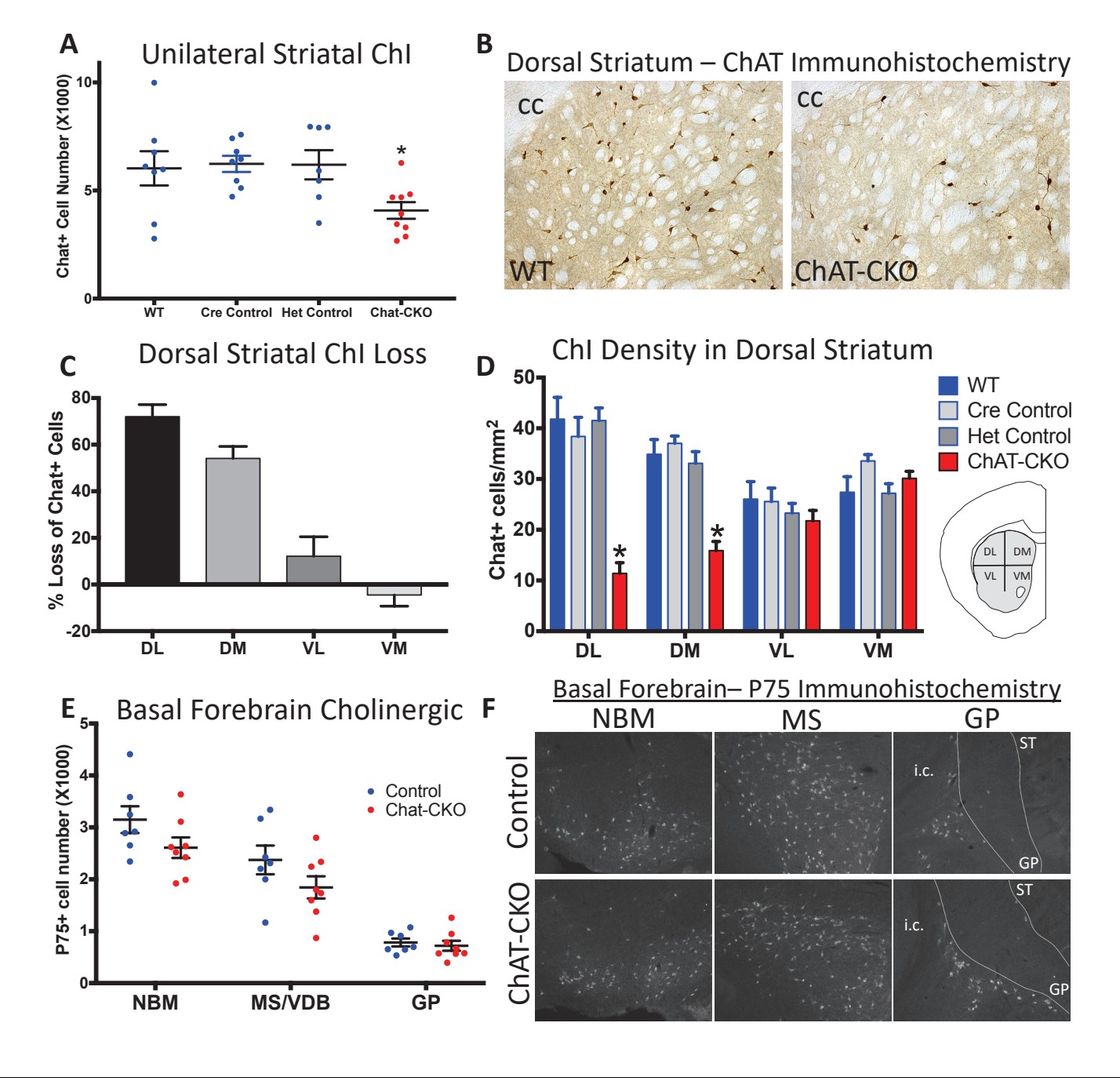

**Figure 1.** Conditional cholinergic neuron deletion of torsinA causes cell autonomous loss of striatal cholinergic neurons. (**A**) Unilateral stereological quantification of the number of ChAT-positive neurons in the striatum of ChAT-CKO and control mice (One-way ANOVA $F_{(3,28)}$ = 3.589, p=0.02, Dunnett's multiple comparisons test: adjusted p value = 0.049; 'WT'=$Tor1a^{Flx/+}$; 'Cre Control'=ChAT-Cre+, $Tor1a^{Flx/+}$; 'Het Control'=$Tor1 a^{Flx/-}$; 'ChAT-CKO'=ChAT-Cre+, $Tor1a^{Flx/-}$). (**B**) ChAT immunohistochemistry of coronal sections containing dorsal striatum from WT and ChAT-CKO mice (cc = corpus callosum). (**C**) Percent reduction in cell density by striatal quadrant (DL = dorsolateral; DM = dorsomedial, VL = ventrolateral, VM = ventromedial). (**D**) Significant ChI loss is selective for dorsal striatal quadrants. Cell density quantification in control and ChAT-CKO striatal quadrants (Two-way ANOVA main effect of genotype $F_{(3,112)}$ = 24.02, p<0.0001; main effect of quadrant $F_{(3,112)}$=8.398, p<0.0001; interaction $F_{(9,112)}$=8.11, p<0.0001. Post-hoc Tukey's multiple comparisons test). (**E**) Basal forebrain neurons are spared in ChAT-CKO mice. Stereological quantification of P75-immunoreactive basal forebrain cholinergic neurons in the nucleus basalis of meynert (NBM), medial septum/nucleus of the vertical limb of the diagonal band (MS/VDB), and globus pallidus (GP). No differences in the number of cholinergic neurons was observed (NBM,

*Figure 1 continued on next page*

*Figure 1 continued*

$t_{(13)}$=1.684, p=0.11; MS/VDB, $t_{(13)}$=1.537, p=0.148; GP, $t_{(13)}$=0.5, p=0.625). (**F**) P75 immunohistochemistry of sagittal sections containing basal forebrain cholinergic neuron populations. i.c. = internal capsule, ST = striatum.

DOI: https://doi.org/10.7554/eLife.36691.002

The following figure supplements are available for figure 1:

**Figure supplement 1.** ChAT-Cre is expressed prenatally.

DOI: https://doi.org/10.7554/eLife.36691.003

**Figure supplement 2.** Independent cohort confirmation of selective striatal cholinergic neuron loss in ChAT-CKO mice.

DOI: https://doi.org/10.7554/eLife.36691.004

**Figure supplement 3.** ChAT-positive neurons are reduced in a topographic pattern throughout the rostrocaudal extent of the dorsal striatum.

DOI: https://doi.org/10.7554/eLife.36691.005

**Figure supplement 4.** Time course of torsinA protein loss in dorsal and ventral striatum.

DOI: https://doi.org/10.7554/eLife.36691.006

**Figure supplement 5.** Time course of torsinA protein loss in basal forebrain.

DOI: https://doi.org/10.7554/eLife.36691.007

neurons exhibited 49% of control torsinA immunoreactivity (*Figure 1—figure supplement 5*). These findings demonstrate that a more rapid loss of torsinA during development does not contribute to the unique vulnerability of dorsal ChI.

TorsinA deletion is restricted to forebrain structures in Dlx-CKO mice. In contrast, ChAT-CKO mice lack torsinA in all cholinergic neurons throughout the neuraxis, enabling us to assess the impact of torsinA loss in additional cholinergic populations. Unbiased stereology of ChAT-immunoreactive neurons in the brainstem demonstrates significantly fewer cholinergic neurons in the pedunculopontine (PPN) and laterodorsal tegmental (LDT) nuclei in 1 year old Chat-CKO mice (*Figure 2A–D*). The PPN and LDT also contain GABAergic, and glutamatergic neurons (*Mena-Segovia, 2016*), which significantly outnumber cholinergic neurons (*Mena-Segovia et al., 2009*; *Wang and Morales, 2009*). Unbiased stereology of NeuN +neurons in PPN and LDT showed no significant change in the overall number of neurons (*Figure 2A,C*). Because cholinergic neurons are a minority of cells in the PPN and LDT, it is possible that a significant reduction of this small sub-population cannot be detected when assessed by counting overall NeuN +neuron number. It is also possible that PPN and LDT cholinergic neurons exhibit reduced ChAT expression rather than actual cell loss. Regardless, either possibility demonstrates a cell autonomous role for torsinA for normal function of these cells. These findings also indicate that the loss or dysfunction of brainstem cholinergic neurons does not have deleterious effects on the viability of surrounding neurons. Consistent with this finding, there was no evidence of reactive microgliosis or astrogliosis in the brainstem (*Figure 2—figure supplement 1*). Quantification of the number of spinal motor neurons (C3-C5; [*Kim et al., 2017*]) demonstrated significantly fewer motor neurons in ChAT-CKO mice (*Figure 2E,F*).

The identification of cholinergic dysfunction or loss in PPN and LDT is notable, as considerable data implicate these cells in motor and postural control. PPN and LDT cholinergic neurons are distributed in a rostrocaudal continuum in the brainstem, forming a coordinated functional unit (*Mena-Segovia, 2016*; *Mena-Segovia and Bolam, 2017*). PPN and LDT cholinergic neurons topographically innervate the striatum and striatal-projecting thalamic and midbrain dopamine neurons (*Dautan et al., 2014*), such that rostral PPN modulates motor-related circuits, LDT innervates limbic circuits, and caudal PPN targets both regions (*Mena-Segovia, 2016*; *Xiao et al., 2016*) via both direct and indirect inputs. Consistent with a central role in modulating locomotor activity, optogenetic stimulation of PPN cholinergic neurons alters locomotion speed, while stimulation of adjacent glutamatergic neurons induces locomotion (*Xiao et al., 2016*; *Roseberry et al., 2016*; *Capelli et al., 2017*). Cholinergic PPN lesion alone or in combination with dopaminergic denervation impairs gait and causes postural abnormalities in primates (*Grabli et al., 2013*; *Karachi et al., 2010*). In rodents, cholinergic-selective PPN lesion impairs performance on the accelerating rotarod and alters sensori-motor gating (*MacLaren et al., 2014a*; *MacLaren et al., 2014b*), while non-specific PPN ablation alters gait (*Blanco-Lezcano et al., 2017*) and impairs reversal learning (*Syed et al., 2016*). Human neuroimaging and postmortem studies also provide support for a connection between PPN cholinergic integrity and motor function. PPN cholinergic loss is linked to gait abnormalities in Parkinson disease (*Karachi et al., 2010*; *Bohnen et al., 2009*), and brainstem lesions (including PPN loss) can

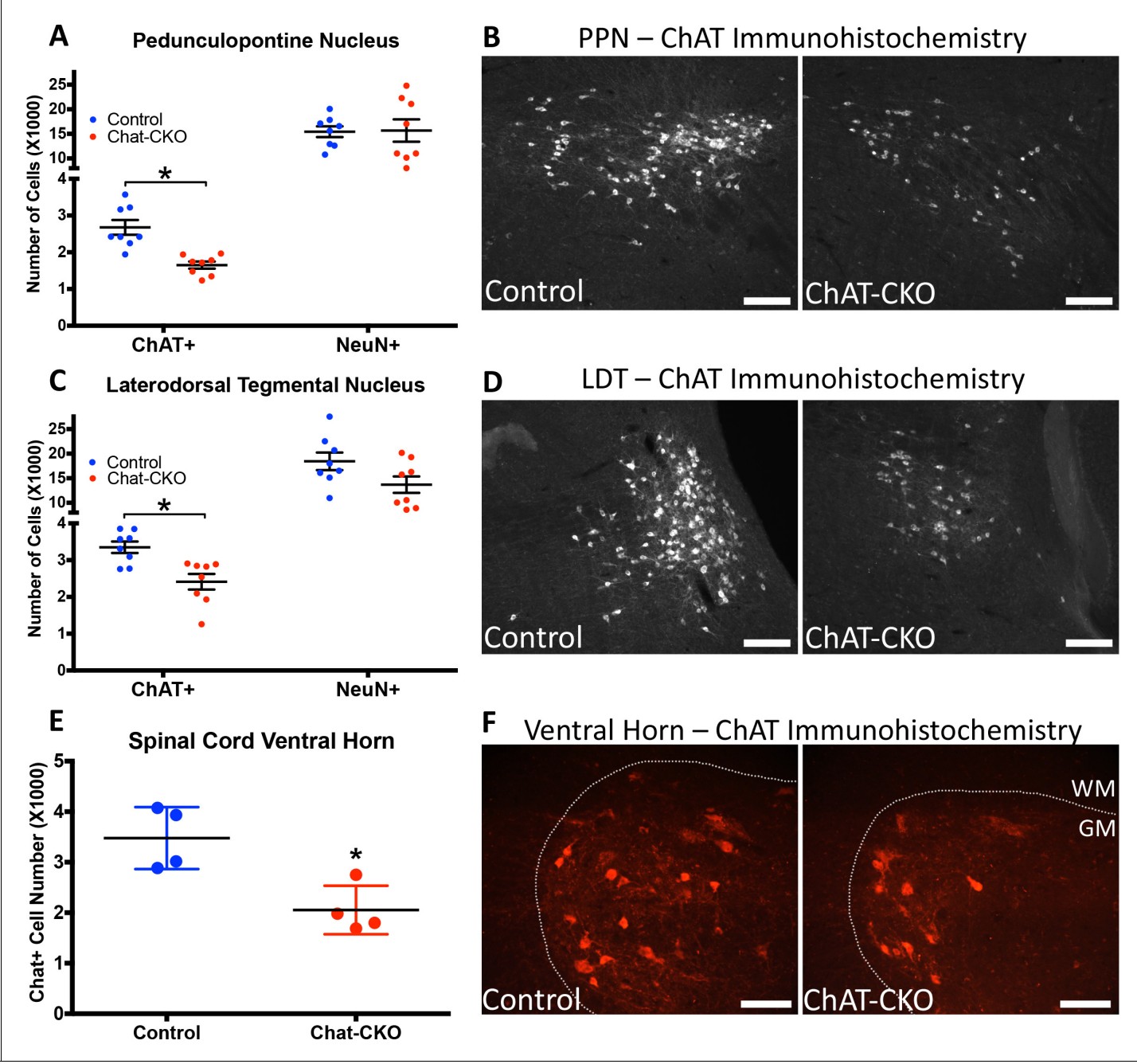

**Figure 2.** ChAT-CKO mice have significantly fewer brainstem and spinal cord cholinergic neurons. (A,B) Stereological quantification of ChAT-positive or NeuN-positive neurons in the pedunculopontine nucleus (PPN) of control and ChAT-CKO mice (ChAT; $t_{(14)}$=4.531, p=0.0005. NeuN; $t_{(14)}$=0.095, p=0.92). (C,D) Stereological quantification of ChAT-positive or NeuN-positive neurons in the laterdorsal tegmental nucleus (LDT) of control and ChAT-CKO mice (ChAT; $t_{(14)}$=3.571, p=0.003. NeuN; $t_{(14)}$=1.934, p=0.073). (E,F) Quantification of the number of ChAT-positive neurons in the cervical spinal cord of control and ChAT-CKO mice ($t_{(6)}$=3.654, p=0.0107). Scale bars = 100 μm.

DOI: https://doi.org/10.7554/eLife.36691.008

The following figure supplement is available for figure 2:

**Figure supplement 1.** Absence of gliosis in the brainstem of ChAT-CKO mice.

DOI: https://doi.org/10.7554/eLife.36691.009

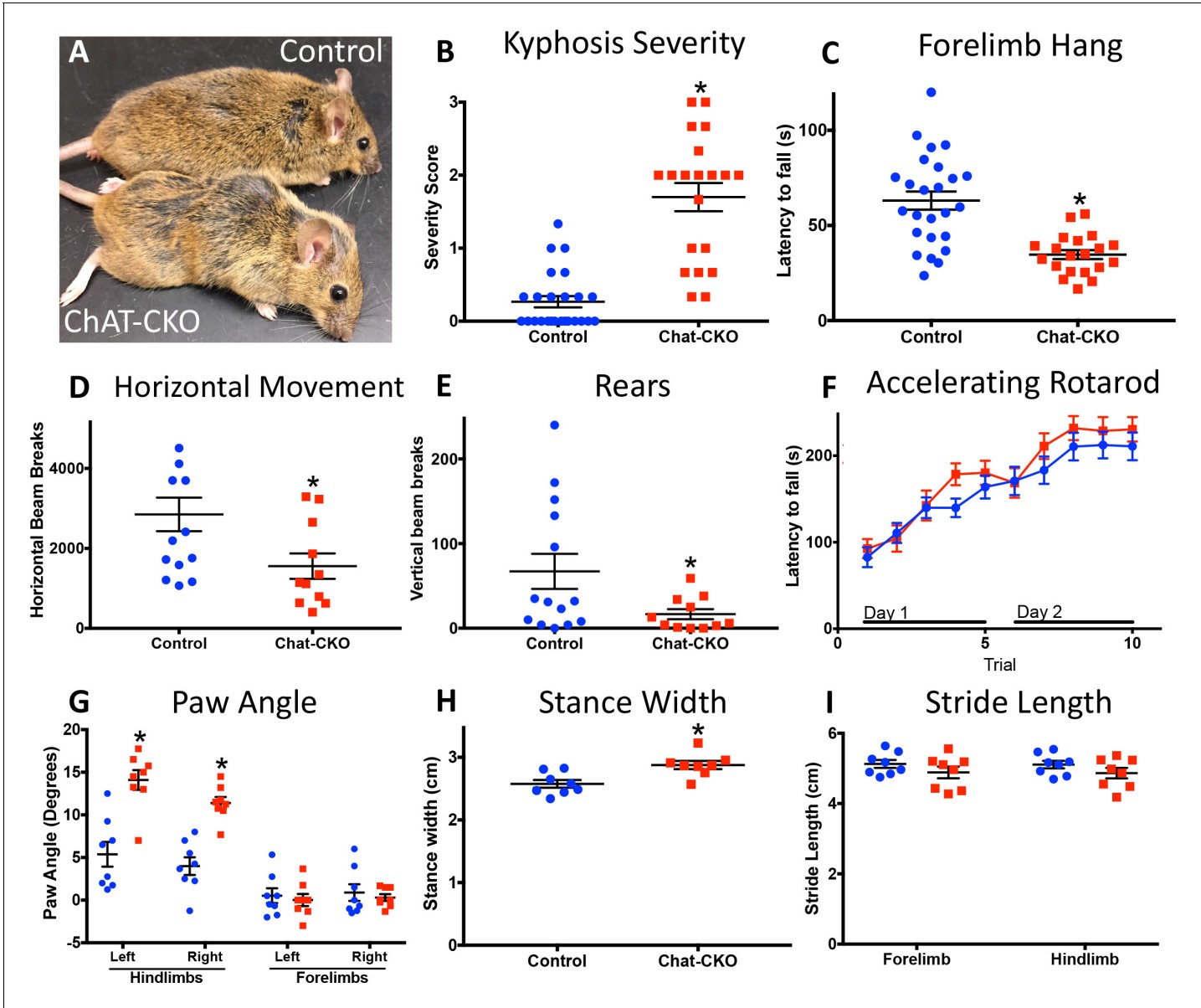

**Figure 3.** Motor behavior is severely disrupted in ChAT-CKO mice. (**A**) Representative image of a control and ChAT-CKO mouse demonstrates severe kyphosis and unkempt coat. (**B**) ChAT-CKO mice exhibit significantly increased kyphotic curvature during locomotion (Mann-Whitney U = 35, p<0.0001). (**C**) ChAT-CKO mice exhibit a significantly reduced latency to fall during forelimb suspension (Mann-Whitney U = 71.5, p<0.0001). (**D, E**) ChAT-CKO mice are hypoactive in the open field (horizontal movement, $t_{(23)}$=2.345, p=0.028; vertical rears, welch-corrected $t_{(15.1)}$ = 2.345, p=0.033). (**F**) Performance on the accelerated rotarod does not significantly differ from controls (two-way repeated measures ANOVA, genotype, $F_{(1,43)}$=0.75, p=0.389; trial, $F_{(9,387)}$=55.63, p<0.0001; interaction, $F_{(9,387)}$=1.194, p=0.297). (**G - I**) ChAT-CKO mouse gait is abnormal during locomotion (*paw angle*, two-way ANOVA main effect of genotype, $F_{(1,56)}$=30.54, p<0.0001, main effect of limb $F_{(3,56)}$=51.02, p<0.0001, interaction $F_{(3,56)}$=13.51, p<0.0001, post-hoc Sidak's multiple comparisons test. *Stance width*, $t_{(14)}$=3.329, p=0.005. *Stride length*, two-way ANOVA genotype $F_{(1,28)}$=3.164, p=0.086, limb $F_{(1,28)}$=0.02, p=0.887, interaction $F_{(1,28)}$=0.0001, p=0.989).

DOI: https://doi.org/10.7554/eLife.36691.010

The following video and figure supplements are available for figure 3:

**Figure supplement 1.** Representative examples of control and ChAT-CKO spinal cords demonstrate significant kyphotic curvature.
DOI: https://doi.org/10.7554/eLife.36691.011
**Figure supplement 2.** ChAT-CKO mice are significantly hypoactive.
DOI: https://doi.org/10.7554/eLife.36691.012
**Figure 3—video 1.** Representative video demonstrating tremulousness, kyphosis, and hyperactivity in ChAT-CKO mice, as compared to controls.
DOI: https://doi.org/10.7554/eLife.36691.013

*Figure 3 continued on next page*

*Figure 3 continued*
**Figure 3—video 2.** ChAT-CKO exhibit twisting and tremulousness, but not limb clasping during tail suspension.
DOI: https://doi.org/10.7554/eLife.36691.014

result in complex dystonia (*Jankovic and Patel, 1983*; *LeDoux and Brady, 2003*; *Loher and Krauss, 2009*; *Zweig et al., 1988*; *Mente et al., 2018*). Systematic cholinergic brainstem cell counts have not been performed in DYT1 dystonia postmortem samples; most studies have failed to demonstrate neuronal inclusions or overt cell loss in this region (*Paudel et al., 2014*; *Pratt et al., 2016*; *McNaught et al., 2004*).

Motor behavior is severely disrupted in ChAT-CKO mice, but is distinct from the Dlx-CKO phenotype (*Figure 3*; *Table 1*). ChAT-CKO pups are initially indistinguishable from littermates, but at approximately 4 weeks of age develop a hunched posture, have unkempt fur, and exhibit reduced responsiveness to handling (*Figure 3A*, *Figure 3—figure supplement 1*). Whereas normal mice exhibit a slight dorsal spinal curvature at rest, ChAT-CKO mice exhibit severe kyphosis, including during locomotion (assessed by two observers blind to experimental conditions; *Figure 3B*; *Figure 3—figure supplement 1*; *Figure 3—video 1*) (*Guyenet et al., 2010*). ChAT-CKO mice also exhibit signs of weakness, including a significantly reduced ability to hang by the forelimbs (*Figure 3C*), tremulous movements, labored breathing (*Figure 3—video 1*), and significantly reduced horizontal and vertical movement in the open field (*Figure 3D,E*, *Figure 3—figure supplement 2*). Remarkably, performance on the accelerating rotarod during two days of training appears normal (*Figure 3F*). The normal rotarod behavior differs from models of motor neuron and neuromuscular disease, suggesting that neuromuscular weakness is modest in ChAT-CKO, and less likely to contribute to other behavioral phenotypes (e.g., postural abnormality). The gait of ChAT-CKO mice is also significantly altered (*Figure 3G–I*). This constellation of behavioral phenotypes is distinct from Dlx-CKO mice (*Table 1*), in which loss of dorsal striatal ChI is associated with a set of persistent abnormal action-induced motor behaviors, including limb clasping and trunk twisting during tail suspension and open field hyperactivity (*Pappas et al., 2015*). ChAT-CKO mice did not exhibit fore- or hindlimb clasping during tail suspension, but did exhibit tremulousness and trunk twisting (15 CKO, 19 heterozygous, 22 Cre control, and 19 wild type mice observed; *Figure 3—video 2*). These results suggest that dorsal striatal ChI neurodegeneration may not, by itself, be sufficient to cause limb clasping during tail suspension. However, the co-occurrence of brainstem and spinal cord

**Table 1.** Behavioral properties of Dlx-CKO and ChAT-CKO mice.

| Motor function | Dlx-CKO | ChAT-CKO |
|---|---|---|
| | *Pappas et al., 2015* eLife 4:e08352 | present manuscript |
| Tail suspension | Trunk twisting | Trunk twisting |
| | Forelimb clasping | - |
| | Hindlimb clasping | - |
| | - | Tremulousness |
| Open field | Hyperactivity | Hypoactivity |
| Rotarod | Normal | Normal |
| Response to handling | Exaggerated | Reduced |
| Weakness, latency to fall | Grid hang reduction | Wire hang reduction |
| Gait | Normal by eye | Abnormal by eye |
| | Slightly reduced stance width | Increased stance width |
| | - | Increased paw angle |
| Overt postural abnormalities | - | Severe kyphosis |
| Tremulous movement | - | Severe |
| Labored breathing | - | Severe |

DOI: https://doi.org/10.7554/eLife.36691.015

**Table 2.** Vulnerability of cholinergic populations.
(*)=Unconfirmed by independent marker.

| Cholinergic population | Cre expression | | Cell death vulnerability | |
|---|---|---|---|---|
| | Dlx-Cre | ChAT-Cre | Dlx-Cre | ChAT-Cre |
| Dorsolateral striatum (including dorsal caudate putamen) | Confirmed | Confirmed | Severe | Severe |
| Dorsomedial striatum (including ventral caudate putamen) | Confirmed | Confirmed | Mild | Spared |
| Nucleus accumbens | Confirmed | Confirmed | - | - |
| Basal forebrain | Confirmed | Confirmed | Spared | Spared |
| Cholinergic Brainstem | Absent | Confirmed | n/a | Severe (*) |
| Primary Motor Neurons | Absent | Confirmed | n/a | Moderate |

DOI: https://doi.org/10.7554/eLife.36691.016

**Table 3.** Properties of cholinergic neuronal populations.
'Nucleus Basalis Complex'=Nucleus Basalis of Meynert, Horizontal limb of the diagonal band of Broca, Ventral Pallidum, Magnocellular Preoptic Area, Substantia Inominata, Nucleus of the Ansa Lenticularis. 'Septa"l = Medial Septum, Vertical Limb of the Diagonal Band of Broca. 'Cholinergic Brainstem'=Pedunculopontine Nucleus, Laterodorsal Tegmental Nucleus (*Pappas et al., 2015*; *Mena-Segovia and Bolam, 2017*; *Gonzales and Smith, 2015*; *Manns et al., 2000*; *Unal et al., 2012*; *Petzold et al., 2015*; *Kanning et al., 2010*; *Kreitzer, 2009*; *Zaborszky et al., 2012*; *Garcia-Rill, 1991*; *Semba et al., 1988*; *Semba and Fibiger, 1992*; *Phelps et al., 1990a*; *Phelps et al., 1988*; *Phelps et al., 1990b*; *Phelps et al., 1989*; *Aroca and Puelles, 2005*; *Schambra et al., 1989*).

| Cholinergic population | Neuronal class | Firing properties | Efferent projections | Afferent inputs | Birth date/ final mitosis | Embryonic origin | ChAT expression |
|---|---|---|---|---|---|---|---|
| Dorsolateral striatum (including dorsal caudate putamen) | Interneuron | tonically active, 2–10 Hz baseline firing rate | Local - striatal spiny projection neurons and fast spiking interneurons | Thalamus, sensorimotor cortex, striatal spiny projection neurons, striatal interneurons | E12-E15 | MGE | ~E16 |
| Dorsomedial striatum (including ventral caudate putamen) | Interneuron | tonically active, 2–10 Hz baseline firing rate | Local - striatal spiny projection neurons and fast spiking interneurons | Thalamus, association cortices, striatal spiny projection neurons, striatal interneurons | E12-E15 | MGE | ~E16 |
| Nucleus accumbens | Interneuron | tonically active, 0.6–12 Hz baseline firing rate | Local - striatal spiny projection neurons and fast spiking interneurons | Thalamus, frontal cortex, striatal spiny projection neurons, striatal interneurons | E12-E15 | MGE | ~E16 |
| Basal forebrain | Projection neuron | Tonic/burst, subtype dependent | Cortex (Nucleus Basalis Complex), Hippocampus (Septal) | Medulla, locus ceruleus, substantia nigra, ventral tegmental area, hypothalamic nuclei, nucleus accumbens, amygdala, local intrinsic GABAergic and glutamatergic collaterals | E11-E15 | POA/MGE | ~E15-16 |
| Cholinergic Brainstem | Projection neuron | episodic | Midbrain, superior colliculus, thalamus, globus pallidus, hypothalamus, septum, striatum, cortex | Brainstem reticular formation, midbrain central gray, lateral hypothalamus-zona incerta, cortex, amygdala, basal forebrain, basal ganglia output nuclei, brainstem and spinal cord sensory nuclei | E12-E13 | Ventral rhombomere 1 (r1) | |
| Primary Motor Neurons | Projection neuron | subtype dependent | Muscle | Motor Cortex, local spinal cord interneurons and sensory neurons | E11-E12 | Ventral spinal cord progenitor domains | E13 |

DOI: https://doi.org/10.7554/eLife.36691.017

neurodegeneration and tremulousness in ChAT-CKO mice could modify a clasping phenotype and therefore limit this strength of this conclusion.

While no single system or experimental approach can fully model a disease, the extreme postural abnormalities (kyphosis and twisting) in ChAT-CKO mice are reminiscent of Oppenheim's original description of dystonia (*Klein and Fahn, 2013*), suggesting that a constellation of cholinergic abnormalities may contribute to such a phenotype. The abnormal gait, tremulous movement, weakness, labored breathing, and appearance of reduced muscle mass in ChAT-CKO mice are consistent with brainstem and spinal cord pathology, yet the time course of ChAT-CKO abnormalities (beginning during development) differ from motor neuron and neuromuscular disease models, in which behavioral phenotypes typically emerge in adulthood (9–11 months of age; (*Dickinson and Meikle, 1973*; *Bridges et al., 1992*; *Deconinck et al., 1997*; *Grady et al., 1997*; *Laws and Hoey, 2004*; *Liu et al., 2016*; *Sopher et al., 2004*; *Monks et al., 2007*). Early motor behavioral manifestations also occur in Dlx-CKO and other DYT1 models, emphasizing the importance of torsinA function during development and maturation at behavioral (*Pappas et al., 2015*; *Liang et al., 2014*), cellular (*Pappas et al., 2018*), and molecular levels (*Tanabe et al., 2016*).

These findings establish a cell autonomous requirement of torsinA for the normal function and survival of distinct populations of cholinergic neurons. Comparison of basic cellular properties between susceptible and invulnerable cholinergic neuron populations does not identify obvious patterns driving selective vulnerability (*Tables 2* and *3*). Within the striatum, dorsal ChI are highly vulnerable to cell death, while ventral ChI are spared. It is unclear whether molecular differences within different ChI populations drive vulnerability, or if differences in connectivity or response to inputs contributes to their loss; these possibilities are not mutually exclusive. While often considered a single neuronal class, an existing and enlarging literature demonstrates that dorsal and ventral striatal ChI exhibit significant differences in morphology, regulation, and receptor expression (reviewed in [*Gonzales and Smith, 2015*]), as well as differing firing patterns during behavioral tasks (*Yarom and Cohen, 2011*) and responses to serotonergic input (*Virk et al., 2016*). These differences implicate the presence of multiple ChI subclasses, though it is important to note that the spared 'ventral' population here represents the ventral part of the dorsal striatum, not the nucleus accumbens. Thalamostriatal and corticostriatal input is highly topographic (*Alexander et al., 1986*; *Smith et al., 2004*), raising the possibility that aberrant input from different thalamic nuclei or cortical regions (or aberrant response to that input) could alter the susceptibility of dorsal vs ventral ChI. It is likely that a combination of these and other factors plays a role in the differential susceptibility of cholinergic neuronal populations, including their molecular profiles (e.g., protective factors in some neurons, susceptibility factors in others), the response to afferent inputs, and their inherent physiological properties.

These studies greatly strengthen the connection between torsinA and cholinergic dysfunction, demonstrating that specific cholinergic populations exhibit a cell autonomous selective vulnerability to torsinA deficiency, while others – basal forebrain and ventral striatum – are spared. These findings open novel avenues of study aimed at defining the molecular mechanisms responsible for this cell autonomous selective vulnerability, and circuit-level analyses to ameliorate the effects of cholinergic neurotransmission abnormalities.

# Materials and methods

**Key resources table**

| Reagent type (species) or resource | Designation | Source or reference | Identifiers | Additional information |
|---|---|---|---|---|
| Gene (Mus musculus) | *Tor1a* | NA | NCBI Gene: 30931; MGI:1353568 | Encodes TorsinA |
| Strain, strain background(M. musculus) | ChAT-Cre | Jackson Laboratories | Stock ID 006410 | Chat[tm2(cre)Lowl]; (Chat-IRES-Cre) |
| Strain, strain background(M. musculus) | Tor1a[Flx/Flx] | Jackson Laboratories | Stock ID 025832 | Tor1a[tm3.1Wtd] |

*Continued on next page*

*Continued*

| Reagent type (species) or resource | Designation | Source or reference | Identifiers | Additional information |
|---|---|---|---|---|
| Strain, strain background(M. musculus) | Tor1a$^{-/-}$ | Jackson Laboratories | Stock ID 006251 | Tor1a$^{tm1Wtd}$ |
| Antibody | Choline Acetyltransferase | Millipore AB144P | RRID: AB_2079751 | 1:100 |
| Antibody | P75 Neurotrophin Receptor | Santa Cruz sc6188 | RRID: AB_2267254 | 1:100 |
| Antibody | NeuN | Cell Signaling #12943 | RRID: AB_2630395 | 1:500 |
| Antibody | GFAP | Cell Signaling #3670P | RRID: AB_561049 | 1:1000 |
| Antibody | Iba-1 | Wako 019–19741 | RRID: AB_839504 | 1:500 |
| Antibody | TorsinA | Abcam ab34540 | RRID: AB_2240792 | 1:100 |
| Antibody | anti-mouse | ThermoFisher A-31571 | RRID: AB_162542 | 1:800 |
| Antibody | anti-rabbit | ThermoFisher A-21206 | RRID: AB_2535792 | 1:800 |
| Antibody | anti-rabbit | ThermoFisher A-31572 | RRID: AB_162543 | 1:800 |
| Antibody | anti-goat | ThermoFisher A-21432 | RRID: AB_2535853 | 1:800 |
| Antibody | anti-goat | Jackson Immunoresearch 705-065-003 | RRID: AB_2340396 | 1:800 |
| Commercial assay or kit | ABC HRP Kit (Standard) | Vector Laboratories | Pk-6100 | Vectastain elite ABC kit |

## Animals

ChAT-CKO mice were generated by crossing *Chat$^{tm2(cre)Lowl}$* mice (*Rossi et al., 2011*) with *Tor1a$^{Flx/Flx}$* mice (*Liang et al., 2014*), using the breeding strategy described in (*Pappas et al., 2015*), and maintained as previously described (*Pappas et al., 2015*).

## Sample size estimation

Sample sizes for histological and behavioral studies were determined by performing a power analysis of the open field or striatal cholinergic stereological data (mean and std. dev.) from (*Pappas et al., 2015*), an alpha of 0.01, and beta of 0.1. (Kane SP. Sample Size Calculator. ClinCalc: http://clincalc. com/stats/samplesize.aspx). Experimental cohorts were generated accordingly.

**Table 4.** Antibodies used for immunohistochemistry.

| Level | Antigen | Host | Conjugated | Dilution | Source |
|---|---|---|---|---|---|
| Primary | Choline Acetyltransferase | Goat | - | 1:100 | Millipore AB144P |
| Primary | P75 Neurotrophin Receptor | Goat | - | 1:100 | Santa Cruz sc6188 |
| Primary | NeuN | Rabbit | - | 1:500 | Cell Signaling #12943 |
| Primary | GFAP | Mouse | - | 1:1000 | Cell Signaling #3670P |
| Primary | Iba-1 | Rabbit | - | 1:500 | Wako 019–19741 |
| Primary | TorsinA | Rabbit | - | 1:100 | Abcam ab34540 |
| Secondary | anti-mouse | Donkey | Alexafluor-647 | 1:800 | ThermoFisher A-31571 |
| Secondary | anti-rabbit | Donkey | Alexafluor-488 | 1:800 | ThermoFisher A-21206 |
| Secondary | anti-rabbit | Donkey | Alexafluor-555 | 1:800 | ThermoFisher A-31572 |
| Secondary | anti-goat | Donkey | Alexafluor-555 | 1:800 | ThermoFisher A-21432 |
| Secondary | anti-goat | Donkey | biotin | 1:800 | Jackson Immunoresearch 705-065-003 |

DOI: https://doi.org/10.7554/eLife.36691.018

**Table 5.** Stereology parameters.

| Region | Marker | Counting frame (μm) | Grid size (μm) | Guard zone (μm) | Dissector (μm) | Section cut thickness (μm) |
|---|---|---|---|---|---|---|
| Striatum | ChAT | 100 × 100 | 250 × 250 | 1 | 10 | 40 |
| NBM | P75 | 90 × 90 | 200 × 200 | 5 | 30 | 50 |
| MS/VDB | P75 | 90 × 90 | 200 × 200 | 5 | 30 | 50 |
| GP | P75 | 100 × 100 | 140 × 140 | 5 | 30 | 50 |
| PPN and LDT | ChAT | 75 × 75 | 150 × 150 | 5 | 30 | 50 |
| PPN and LDT | NeuN | 40 × 40 | 250 × 250 | 5 | 30 | 50 |

DOI: https://doi.org/10.7554/eLife.36691.019

### Imaging and stereology

Brain sections were generated and stained with immunohistochemistry using the methods described in (*Pappas et al., 2015*; *Pappas et al., 2018*). Antibodies and reagents are listed in *Table 4*. Sections were observed with epifluorescence or brightfield microscopy (*Pappas et al., 2018*), and unbiased stereological cell counting was performed with StereoInvestigator software using the Optical Fractionator probe (specific parameters in *Table 5*). Striatal cell density was quantified as done previously (*Pappas et al., 2015*). Spinal cord neurons were quantified as described in (*Kim et al., 2017*).

### Behavioral analysis

Tail suspension, forelimb wire suspension, open field, accelerating rotarod, and gait analysis were performed as described in (*Pappas et al., 2015*). Kyphosis severity was scored as described in (*Guyenet et al., 2010*).

### Statistical analysis

t-tests, one-way, or two-way ANOVA with posthoc corrections for multiple comparisons were performed to compare experimental groups (details in each figure legend). If variances were significantly different between groups, non-parametric tests were performed.

## Acknowledgements

We thank Stephanie Mrowczynski for expert technical assistance and the Dauer lab for helpful comments and suggestions. This research was supported by generous support from Tyler's Hope for a Dystonia Cure and the following grants: RO1NS077730 (William T Dauer), RO1NS057482, R21NS099921, and R56NS104218 (Umrao R Monani).

## Additional information

### Funding

| Funder | Grant reference number | Author |
|---|---|---|
| National Institute of Neurological Disorders and Stroke | RO1NS077730 | William T Dauer |
| Tyler's Hope for a Dystonia Cure | | William T Dauer |
| National Institutes of Health | RO1NS057482 | Umrao R Monani |
| National Institutes of Health | R21NS099921 | Umrao R Monani |
| National Institutes of Health | R56NS104218 | Umrao R Monani |

The funders had no role in study design, data collection and interpretation, or the decision to submit the work for publication.

## Author contributions
Samuel S Pappas, Conceptualization, Data curation, Formal analysis, Investigation, Writing—original draft, Writing—review and editing; Jay Li, Tessa M LeWitt, Formal analysis, Investigation, Writing—review and editing; Jeong-Ki Kim, Investigation, Writing—review and editing; Umrao R Monani, Resources, Supervision, Funding acquisition, Writing—review and editing; William T Dauer, Conceptualization, Resources, Supervision, Funding acquisition, Writing—review and editing

## Author ORCIDs
Samuel S Pappas (iD) http://orcid.org/0000-0002-6980-2058
Jay Li (iD) http://orcid.org/0000-0002-8146-4450
Jeong-Ki Kim (iD) http://orcid.org/0000-0003-0218-1215
William T Dauer (iD) http://orcid.org/0000-0003-1775-7504

## Ethics
Animal experimentation: All experiments were performed according to the recommendations in the Guide for the Care and Use of Laboratory Animals of the National Institutes of Health. All procedures involving animals were approved by the University of Michigan Institutional Animal Care and Use Committee (animal use protocol PRO00006600). All effort was taken to minimize the number of animals used and to prevent discomfort or distress.

## Decision letter and Author response
Decision letter https://doi.org/10.7554/eLife.36691.023
Author response https://doi.org/10.7554/eLife.36691.024

# Additional files

## Supplementary files
• Transparent reporting form
DOI: https://doi.org/10.7554/eLife.36691.020

## Data availability
All data generated during this study are included in the manuscript and supporting files

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
