## [Decision Letter]

Thank you for submitting your article "A cell autonomous torsinA requirement for cholinergic neuron survival and motor control" for consideration by *eLife*. Your article has been reviewed by three peer reviewers, including Louis J Ptáček as the Reviewing Editor and Reviewer #1, and the evaluation has been overseen by a Senior Editor.

The reviewers have discussed the reviews with one another and the Reviewing Editor has drafted this decision to help you prepare a revised submission..

Summary:

In a previous study (Pappas et al., 2015), Pappas et al. made the important findings that deletion of torsinA in embryonic progenitors of forebrain cholinergic and GABAergic neurons caused selective degeneration of dorsal striatal cholinergic interneurons, and that the loss of dorsal striatal cholinergic interneurons was sufficient to cause dystonic-like twisting movements that emerged during juvenile CNS maturation.

The original paper already described selective loss of ChI. Dlx5/6 is far broader in its expression than ChAT Cre – it encompasses all forebrain inhibitory neurons and cholinergic neurons. So it would have been hard to know if the ChI degeneration was due to loss of all other interneurons or if was cell-intrinsic, so here they show it is specific to cholinergic. It also associated motor dysfunction with the ChI loss – and a central point of that paper was that cholinergic cells *were* responsible since motor defects were rescued by cholinergic pharmacology. Given this knowledge, the new data showing torsinA deletion from cholinergic neurons causes motor dysfunction is less interesting. In other work, the authors defined a postnatal developmental window where neurons were most susceptible to torsinA loss. At least, this was the take home message of their study (Tanabe, 2016). Given this, there is some worry about whether the difference in neuronal vulnerability relates to the speed that torsinA is lost after Cre-mediated deletion. We don't know that Cre expression is the same across all ChI, in terms of levels or timing. We also don't know if torsinA half-life is the same in all ChI – and likely is not and may depend on cellular metabolic activity, etc. It doesn't appear that we have a good handle on torsinA half-life in general, and it's potentially quite long lived. Also, what role might torsinB have in reducing vulnerability in some cells? The most important conclusion, that the phenotype is indeed from cholinergic cells and not all inhibitory neurons in striatum, is the most notable part of this manuscript. It is amazing that a very small percentage of neurons in the dorsal striatum can have such dramatic and quite specific effects on behavior in a disease model.

While the data here (and their other papers) are convincing, we feel it is important to completely exclude the possibility that selective neuronal vulnerability (that they focus on here) derives from technical vs. disease-relevant biology. The reviewers wanted to be sure that ChAT-Cre is not expressed earlier or more broadly than expected. The story is more exciting if associated with a satisfactory explanation of differential vulnerability. Also, it is essential to ensure that differential vulnerability is not due to differences in completeness of torsinA deletion in different cell types. Are reagents available to do this in a timely manner? It is important to compare efficiency of deletion in both KOs, although the key finding is that ChAT alone reproduces the broad Dlx5/6 KO. They can do in situ to evaluate effectiveness of KO.

Essential revisions:

1) It is not clear why some populations of cholinergic neurons are more vulnerable to torsinA deficiency. Although this is a hard question to address, inclusion of other prominent cholinergic neurons and contrasting their cellular properties may provide some clues. The manuscript gives the impression that cholinergic neurons related to motor functions are more vulnerable. This seems to be a circular argument (not a mechanistic one).

2) In PPN and LDT, both cholinergic neurons and NeuN+ cells were counted. It is surprising that there was loss of cholinergic neurons but not NeuN+ cells. It is likely that noise in the data diluted the signal in counting NeuN+ cells. However, it is also possible that loss of cholinergic neurons is due to loss of cholinergic marker expression. This needs more data and/or explanation.

3) The motor symptom differences between ChAT-CKO and Dlx-CKO were not explained clearly. It will be helpful to list the main differences between the two manipulations at both the circuit level and behavioral level to give more insights into the phenotypic differences and functional implications.

4) The new information is limited to showing that ChI loss occurs because of torsinA loss in the ChI themselves. This does not answer whether different ChI populations have different molecular profiles that render them more or less sensitive to torsinA loss (so that dorsal vs. ventral ChI might be considered as different cell types). Alternatively, it might still be that the nature of ChI connectivity differs between striatal regions so that only one set of cells requires torsinA (for example if degeneration depends on excitotoxicity, and ChI in dorsal striatum receive more direct glutamatergic inputs).

5) A technical issue is whether it is 100% clear that the genetic technology causes the exact same loss of torsinA in the two populations. The manuscript shows Cre expression across the striatum, but this is not synonymous with a time course of how torsinA protein is lost from the two populations. Is it possible that dorsal neurons are more sensitive because they more rapidly lose torsinA protein? This needs very careful controls given it is central to their conclusion that the regional specificity has physiological relevance.

6) I am also left wondering how ChI degeneration relates to other mechanisms shown by the group like abnormal nuclear pore complexes, or torsinB expression. Do these have any role in the selective vulnerability? Further, is there a link between ChI vulnerability and that of the deep cerebellar nuclei and sensorimotor cortical neurons that they showed are selectively lost when torsinA is deleted across the brain (Liang, 2014).

---

## [Author Response]

Essential revisions:1) It is not clear why some populations of cholinergic neurons are more vulnerable to torsinA deficiency. Although this is a hard question to address, inclusion of other prominent cholinergic neurons and contrasting their cellular properties may provide some clues. The manuscript gives the impression that cholinergic neurons related to motor functions are more vulnerable. This seems to be a circular argument (not a mechanistic one).

We appreciate this comment and have removed all statements implying that connections to motor function render neurons more vulnerable. We also include a new table comparing the cellular properties (neuronal class, firing properties, efferent projections, afferent input, birth dates, embryonic origins, time of first ChAT expression, and vulnerability to cell death) between dorsal striatum, ventral striatum, basal forebrain, PPN/LDT, and primary motor neurons (Table 2). Although obvious patterns between vulnerable populations do not immediately emerge, we believe that this information will be valuable for future studies stimulated by our new findings, aimed at further advancing understanding of the mechanisms of selective vulnerability.

2) In PPN and LDT, both cholinergic neurons and NeuN+ cells were counted. It is surprising that there was loss of cholinergic neurons but not NeuN+ cells. It is likely that noise in the data diluted the signal in counting NeuN+ cells. However, it is also possible that loss of cholinergic neurons is due to loss of cholinergic marker expression. This needs more data and/or explanation.

We are also surprised that NeuN+ cell numbers were not different in the LDT and PPN and agree that there are other potential explanations for the ChAT+ cell numbers, including reduction of ChAT expression without frank cell loss.

Several anatomical studies demonstrate that non-cholinergic cell types in the PPN and LDT greatly outnumber cholinergic neurons. There are at least twice as many GABAergic as cholinergic neurons (Mena-Segovia et al., 2009), and glutamatergic neurons are at least as abundant as GABAergic, or present at even higher numbers (Martinez-Gonzalez et al., 2012; Wang and Morales, 2009). Indeed, in the rostral PPN, the density of GABAergic neurons is more than 5 times higher than cholinergic neurons (Martinez-Gonzalez et al., 2011). Because cholinergic neurons represent a small minority of cells in the PPN and LDT, we believe it most likely that reduction of this small population of cells was ‘lost in the noise’ of the overall NeuN+ cell counts, as suggested.

To specifically address the issue of cell loss versus ChAT downregulation, we attempted to define alternative markers of brainstem cholinergic neuron populations. We performed a series of studies examining P75 and VAChT as additional markers. Unfortunately, these molecules did not reliably or reproducibly mark PPN and LDT cell bodies. P75 was present in synaptic inputs to cholinergic brainstem neurons but not in PPN or LDT neurons themselves. VAChT appeared punctate throughout the region, but cell soma expression was not clear or sufficiently defined for stereological assessment. Unlike striatal ChI, cholinergic brainstem neurons are not uniformly larger than other intermingled cell populations, preventing us from using cell size as a proxy (as previously done for forebrain populations (Pappas et al., 2015)). We therefore cannot provide additional evidence to support the presence of cholinergic brainstem neurodegeneration at this time.

In our revised manuscript, we highlight these valuable new points. We make clear the relative numbers of cholinergic and non-cholinergic cells in these brainstem nuclei and point out that our findings in the brainstem may reflect downregulation of ChAT as opposed to cell loss (as clearly occurs in striatum). Importantly, whether these cells “only” lose ChAT expression, or degenerate, our findings are the first to demonstrate a cell autonomous torsinA requirement for brainstem cholinergic neuron function.

3) The motor symptom differences between ChAT-CKO and Dlx-CKO were not explained clearly. It will be helpful to list the main differences between the two manipulations at both the circuit level and behavioral level to give more insights into the phenotypic differences and functional implications.

We appreciate this feedback. We have now generated a table outlining all known motor features of Dlx-CKO and ChAT-CKO mice (Table 1), which will greatly facilitate for the reader a direct comparison between the models.

4) The new information is limited to showing that ChI loss occurs because of torsinA loss in the ChI themselves. This does not answer whether different ChI populations have different molecular profiles that render them more or less sensitive to torsinA loss (so that dorsal vs. ventral ChI might be considered as different cell types). Alternatively, it might still be that the nature of ChI connectivity differs between striatal regions so that only one set of cells requires torsinA (for example if degeneration depends on excitotoxicity, and ChI in dorsal striatum receive more direct glutamatergic inputs).

We agree that the major mechanistic finding of this paper is that torsinA loss in ChI themselves causes cell death, rather than via non-cell autonomous mechanisms of surrounding torsinA deficient neurons. This result is striking and represents an important advance considering the subregion specificity of the loss (to the most dorsal “motor” aspects of dorsal striatum) is driven by a cell autonomous effect of torsinA in a population of cells typically considered “uniform.” The intriguing and important question raised by the reviewers speaks to the interesting issues raised by our novel finding: do multiple unique ChI cell types exist (one vulnerable, others spared), or do connectivity differences drive differential susceptibility to torsinA loss of function.

There is precedent for either possibility (or both). While often considered a single neuronal class, dorsal and ventral striatal ChI exhibit significant differences in morphology, regulation, and receptor expression (reviewed in (Gonzales and Smith, 2015)), as well as different responses to serotonin input (Virk et al., 2016) and differential firing patterns during some motor behavioral tasks (Yarom and Cohen, 2011). These differences are consistent with the existence of multiple ChI subclasses, though it is important to note that the “ventral” population in our manuscript is the ventral part of the dorsal striatum, not the nucleus accumbens. In our revised manuscript we have better highlighted this literature, which we believe – together with our new findings – will stimulate additional work in this interesting area.

It is also possible that the unique pattern of afferent inputs could contribute to the striking pattern of subregion vulnerability we observe, as thalamostriatal and corticostriatal input is highly topographic (Alexander et al., 1986; Smith et al., 2004).

We think it most likely that a combination of factors plays a role in the differential susceptibility of different cholinergic neuronal populations, including their molecular profiles (e.g., protective factors in some neurons, susceptibility factors in others), the response to differential afferent inputs, and their inherent physiological properties. Our new findings point to future experiments comparing vulnerable and invulnerable populations at multiple mechanistic levels that will be required to elucidate the mechanism(s) responsible differential vulnerability. These important questions will require significant experimental effort – likely years worth of work – which we believe is beyond the scope of this manuscript. However, to help stimulate future work, we have added a short discussion of different possible explanations for differential susceptibility to the Discussion section.

5) A technical issue is whether it is 100% clear that the genetic technology causes the exact same loss of torsinA in the two populations. The manuscript shows Cre expression across the striatum, but this is not synonymous with a time course of how torsinA protein is lost from the two populations. Is it possible that dorsal neurons are more sensitive because they more rapidly lose torsinA protein? This needs very careful controls given it is central to their conclusion that the regional specificity has physiological relevance.

We appreciate the reviewer raising this very important point, which we had not previously systematically examined. To further explore this issue, we assessed torsinA levels in cholinergic neurons at P0 in ChAT-CKO and Cre negative control mice (the same time for which we document uniform Cre expression). Despite unambiguous prenatal Cre expression (Figure 1—figure supplement 1), our new data demonstrate that at least 50% of torsinA remained in cholinergic neurons at P0 using the semi-quantitative measurement of fluorescence intensity. To determine whether, following Cre-recombination, the levels of torsinA loss correspond to cell loss, we compared the levels of torsinA immunofluorescence in dorsal striatal (vulnerable) and ventral striatal (invulnerable) ChI at P0. Surprisingly, the invulnerable ventral striatal ChI exhibited a significantly greater decrease of torsinA (~48% decrement) compared to vulnerable dorsal striatal ChI population (~18% decrement) (Two-way ANOVA with post-hoc Sidak’s multiple comparisons test. ChAT-CKO dorsal vs ventral p<0.0002. Full details in legend to Figure 1—figure supplement 4). We also assessed torsinA levels in basal forebrain cholinergic projection neurons, which are spared in all DYT1 models assessed. Basal forebrain cholinergic neurons from ChAT-CKO mice exhibited an ~51% of loss of torsinA (compared to control; p<0.0001, Welch’s t-test; Figure 1—figure supplement 5), similar to the levels in ventral ChI. Considered together, these data argue against the possibility that more rapid loss of torsinA from dorsal striatal neurons contributes to their selective degeneration. These findings also eliminate the possibility of a technical artifact whereby Cre recombination occurs selectively in dorsal ChI.

6) I am also left wondering how ChI degeneration relates to other mechanisms shown by the group like abnormal nuclear pore complexes, or torsinB expression. Do these have any role in the selective vulnerability? Further, is there a link between ChI vulnerability and that of the deep cerebellar nuclei and sensorimotor cortical neurons that they showed are selectively lost when torsinA is deleted across the brain (Liang, 2014).

We agree that linking previously described torsinA loss-of-function mediated phenotypes (nuclear pore complex abnormalities or ubiquitin accumulation (Liang et al., 2014; Pappas et al., 2018) is of interest and could help to advance a theme underlying selective cell vulnerability. Interestingly, the events occurring in striatal ChI may be distinct from those previously defined (including in DCN and sensorimotor cortex). In contrast to these non-cholinergic neuronal populations, we do not observe nuclear pore complex or ubiquitin abnormalities in striatal ChIs (Pappas et al., 2018). As noted, our prior work a links torsinB levels to the developmental nuclear envelope phenotypes (Kim et al., 2010; Tanabe et al., 2016), but that work did not directly address the relationship between torsinB and cell death. We have used laser capture microdissection to examine the levels of torsinB mRNA in wild type and torsinA null striatal ChI; we find no difference in torsinB levels between these conditions so think it is unlikely to be playing a role in this specific context (we would be happy to include these data if the reviewers deem it essential). As noted in our response to the prior question, these new findings are the first to identify a cell autonomous role for torsinA in a subpopulation of striatal ChI critical for motor function (and which as strongly implicated in the disease). This new finding represents a significant advance in understanding the mechanism of selective ChI loss reported in our original *eLife* publication, setting the stage for future studies that we believe to be beyond the scope of the current manuscript.